# Epidermal growth factor receptor signaling uncouples germ cells from the somatic follicular compartment at ovulation

Laleh Abbassi[1,2,7], Stephany El-Hayek[1,3,8], Karen Freire Carvalho[1,2], Wusu Wang [1,4], Qin Yang[1], Sofia Granados-Aparici[1], Rafael Mondadori [5], Vilceu Bordignon [5] & Hugh J. Clarke [1,2,3,6✉]

Germ cells are physically coupled to somatic support cells of the gonad during differentiation, but this coupling must be disrupted when they are mature, freeing them to participate in fertilization. In mammalian females, coupling occurs via specialized filopodia that project from the ovarian follicular granulosa cells to the oocyte. Here, we show that signaling through the epidermal growth factor receptor (EGFR) in the granulosa, which becomes activated at ovulation, uncouples the germ and somatic cells by triggering a massive and temporally synchronized retraction of the filopodia. Although EGFR signaling triggers meiotic maturation of the oocyte, filopodial retraction is independent of the germ cell state, being regulated solely within the somatic compartment, where it requires ERK-dependent calpain-mediated loss of filopodia-oocyte adhesion followed by Arp2/3-mediated filopodial shortening. By uncovering the mechanism regulating germ-soma uncoupling at ovulation, our results open a path to improving oocyte quality in human and animal reproduction.

[1] Research Institute of the McGill University Health Centre, Montreal, Canada. [2] Division of Experimental Medicine, McGill University, Montreal, Canada. [3] Department of Biology, McGill University, Montreal, Canada. [4] College of Animal Science and Technology, Northwest A&F University, Shaanxi, PR China. [5] Department of Animal Science, McGill University, Montreal, Canada. [6] Department of Obstetrics and Gynecology, McGill University, Montreal, Canada. [7] Present address: Princess Margaret Cancer Centre, University Health Network, Toronto, Canada. [8] Present address: Centre for Arab Genomic Studies, Dubai, United Arab Emirates. ✉email: Hugh.clarke@mcgill.ca

Cell contact and adhesion play indispensable roles in the development of multicellular organisms and maintenance of tissue homeostasis[1], and the loss of intercellular adhesion is frequently associated with pathological events including metastatic migration of tumor cells[2]. Germ cells, however, present a rare and striking exception to this general principle. Like most other cell types, their differentiation requires intimate contact and interaction with supporting somatic cells of the gonad, which provide essential molecular nutrients and regulatory signals[3–9]. In contrast to most other cell types, however, once germ cells are mature, their function requires loss of contact with the somatic support cells, freeing them to participate in fertilization. The timely loss of germ-soma contact is linked to improved gamete and embryo quality including in humans[10–12], indicating that mechanisms have evolved to ensure that germ-somatic contact is not disrupted precociously when the germ cells are immature, yet is disrupted efficiently in mature gametes so that they may interact to generate a new organism. Despite the functional imperative of breaking germ-soma contact prior to fertilization, little is known in any species of the mechanisms that underpin this critical process.

The mammalian follicle represents an ideal model system to address this gap in knowledge. First, its architecture is simple and well defined. The centrally located oocyte is surrounded by concentric layers of somatic granulosa cells, a basement membrane, and theca cells[13]. Interposed between the growing oocyte and the granulosa cells is a thick extracellular matrix, termed the zona pellucida[14]. Overcoming this physical impediment, germ-soma contact and communication is mediated by specialized actin-rich filopodia, termed transzonal projections (TZPs), that extend from the granulosa cells through the zona pellucida to the oocyte[13,15–18]. Thus, the physical structures mediating germ-soma adhesion are clearly defined and can be easily visualized. Second, adhesion is broken at a defined time in the reproductive cycle. Luteinizing hormone (LH) released by the anterior pituitary at the time of ovulation binds to its receptor (LHCGR; luteinizing hormone/choriogonadotropin receptor) in the mural granulosa cells adjacent to the follicular wall, triggering among other events the release of epidermal growth factor (EGF)-like peptides that activate the EGF receptor (EGFR) located on both the mural granulosa and the cumulus granulosa adjacent to the oocyte (Fig. 1)[19–21]. Although the relative roles of direct LHCGR and EGFR-mediated signaling remain to be fully established, this signaling cascade triggers the final phase of oocyte differentiation, termed meiotic maturation. Germ-soma contact is irreversibly broken at this time, as the TZPs disappear and the cumulus granulosa cells become displaced away from the oocyte. Third, conditions have been developed for culturing complexes comprising the oocyte and the surrounding somatic follicular cells. These conditions maintain the natural 3-dimensional architectural relationship between the two cell types and support both growth and maturation, thus enabling experimental analysis of

the underlying mechanisms in a physiologically relevant context[22–26].

Here we show that administration of an LH analogue in vivo or EGF in vitro triggers a loss of the TZPs that occurs uniformly 4–8 h later. TZP loss is not due to displacement of cumulus cells away from the oocyte, but rather to an active and highly synchronized retraction of these processes into the cumulus cell bodies. Unexpectedly, and in contrast to the process of elaboration, TZP retraction is not regulated by oocyte-derived signals, but is instead controlled exclusively by signaling within the granulosa cells. TZP retraction depends on ERK-dependent activation in these cells of calpain, which triggers loss of adhesion between the TZPs and the oocyte, and on the actin-assembly nucleator complex, Arp2/3, whose activity is required for the TZPs to shorten. We further show that granulosa cells do not acquire the ability to undergo TZP retraction until the antral stage of folliculogenesis, indicating that it is developmentally regulated. Our results uncover mechanistic elements of the process that frees the germ cell from control by its somatic microenvironment at the time of ovulation, enabling it to participate in fertilization.

## Results

**Activation of EGFR signalling induces retraction of TZPs.** To uncover the mechanism responsible for TZP loss, we first established its timing. We injected female mice with the FSH analogue, equine chorionic gonadotropin (eCG), to promote the final stage of follicular development, followed by the LH analogue, human chorionic gonadotropin (hCG), to induce maturation. At different times following hCG injection, we isolated cumulus-oocyte complexes (COCs) from antral follicles, fixed them, and stained them using phalloidin, a bicyclic peptide originally isolated from *Amanita phalloides* that binds to filamentous (F-) actin, to label TZPs and DAPI to label DNA.

Prior to the initiation of maturation, a dense network of TZPs links the granulosa cells to the immature oocyte, whose chromatin is dispersed within the nucleus, termed the germinal vesicle (GV) (Fig. 2a, 0 h, arrow indicates TZPs). By 4 h post-hCG, the maturing oocytes had undergone germinal vesicle breakdown (GVBD) although the condensed chromosomes had not yet become aligned on the first meiotic spindle (Fig. 2a, 4 h, inset). At this time, the TZP network remained intact (Fig. 2a; Fig. 2b, 100.0 ± 0.25% [mean ± standard error of the mean] $n = 12$), consistent with reports that gap junctional communication between the two cell types persists for a short period of time after GVBD[10,27]. Strikingly, however, by 8 h post-hCG virtually no TZPs could be detected (Fig. 2a, 8 h; Fig. 2b, 0.03 ± 0.007%, $n = 11$). These results indicate, first, that TZPs are lost approximately midway through maturation and, second, that the loss occurs several hours after the maturing oocyte has undergone GVBD. Because the subset of follicles that have responded to the FSH activity by expressing LH-receptors is challenging to reliably

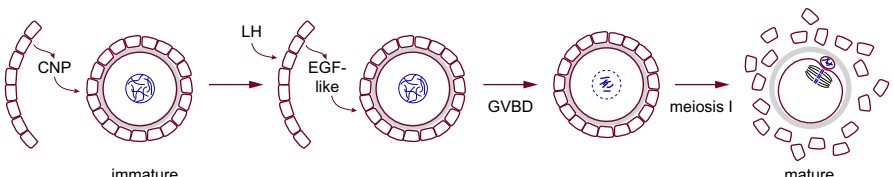

**Fig. 1 Schematic of maturation.** The oocyte is surrounded by cumulus granulosa cells (one layer only shown for simplicity). Prior to maturation, C-type natriuretic peptide (CNP) produced by the mural granulosa cells of antral follicles maintains the immature state. At ovulation, LH binds to receptors on mural granulosa cells, triggering release of EGF-like peptides. These bind to EGFR on both the mural and the cumulus granulosa. LH and EGFR activity cooperatively trigger maturation, marked by germinal vesicle breakdown (GVBD), completion of the first meiotic division and alignment of the oocyte chromosomes on the metaphase II spindle. During maturation, the cumulus granulosa cells become displaced away from the oocyte.

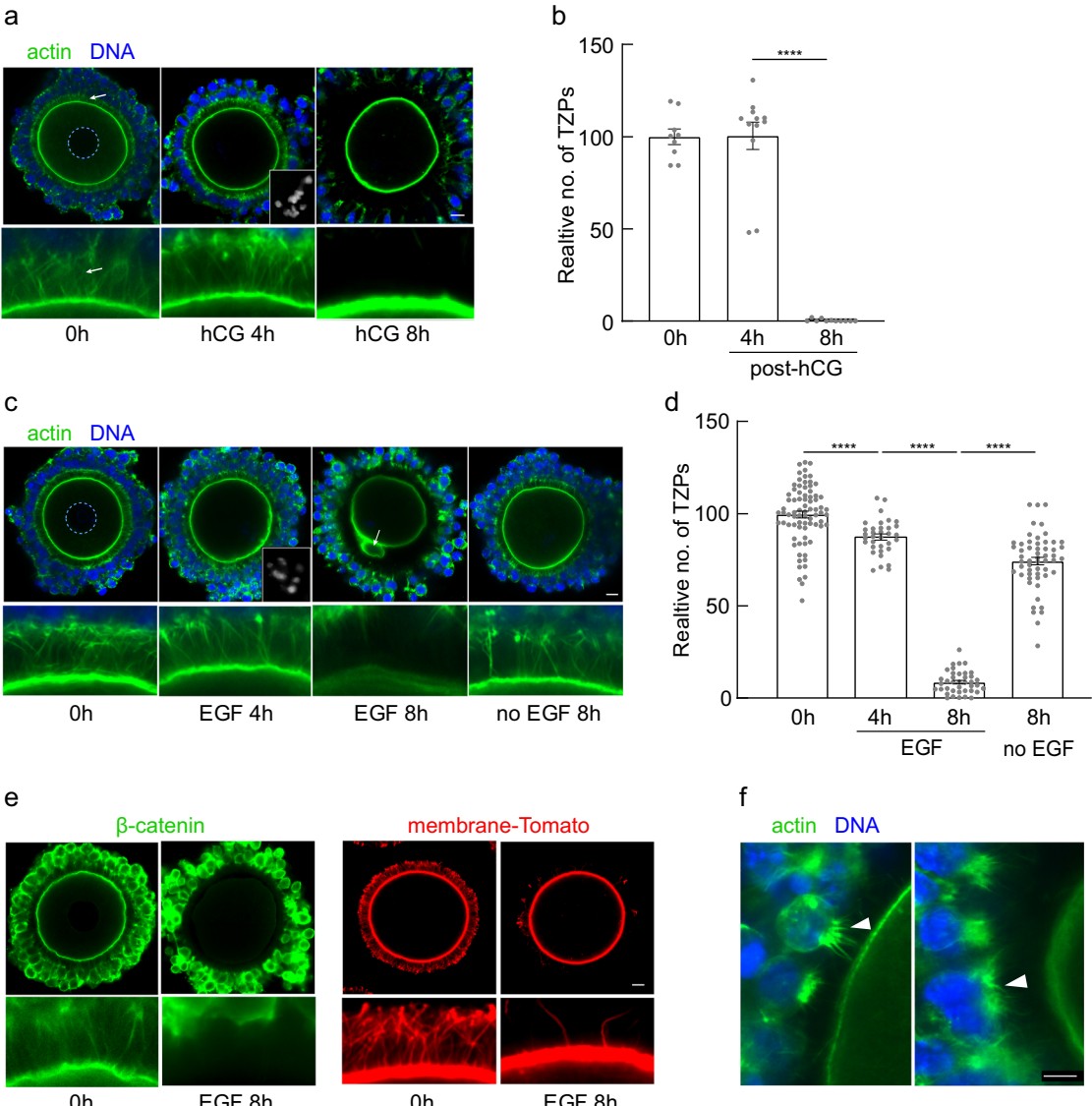

**Fig. 2 EGF triggers retraction of TZPs. a** Confocal images of cumulus-oocyte complexes (COCs) during maturation in vivo. TZPs (arrows) are stained using the F-actin binding peptide, phalloidin. Lower panels show enlarged portion of corresponding upper panels. 0 h: Prior to maturation. Dashed circle outlines the oocyte nucleus (germinal vesicle). 4 h: 4 h after injection of hCG, an LH analogue. Inset shows condensed oocyte chromosomes, confirming that GVBD has occurred. 8 h: 8 h post-hCG. **b** Mean and SEM of number of TZPs at the indicated times after hCG injection, normalized to the number at 0 h. Each point represents an individual COC. $n = 9$ (0 h), 12 (4 h), 11 (8 h) COCs examined over three independent experiments. Statistical analysis using one-way ANOVA with Tukey multiple-comparison test. **** $P < 0.0001$. **c** Confocal images of COCs obtained from antral follicles of eCG-primed mice and incubated for the indicated periods of time in the presence of EGF or in its absence. Inset in EGF 4 h panel shows condensed chromosomes, confirming GVBD. Arrow in EGF 8 h panel shows a polar body, confirming completion of the 1st meiotic division. **d** Mean and SEM of number of TZPs at the indicated times after addition of EGF, normalized to the number at 0 h. Each point represents an individual COC. $n = 77$ (0 h), 34 (4 h EGF), 39 (8 h EGF) 55 (8 h no EGF) COCs examined over five independent experiments. Statistical analysis using one-way ANOVA with Tukey multiple-comparison test. **** $P < 0.0001$. **e** Confocal images of COCs of wild-type mice stained at the indicated times using anti-β-catenin or mTmG mice stained using anti-RFP to detect the cell membrane. Cumulus cells were removed prior to staining using anti-RFP to enable the signal in the TZPs to be detected. **f** Retracting TZPs cluster to form a bouquet of actin (arrowheads) at the oocyte-facing side of the granulosa cells. Scale bar = 10 μm (**a**, **c**, **e**); 5 μm (**f**).

identify in intact ovaries, it was not feasible to use in vivo approaches to further dissect the timing and mechanism of TZP loss. We therefore turned to an established in vitro model whereby COCs are isolated from large antral follicles of eCG-primed mice and allowed to mature in vitro[21].

The cumulus granulosa cells, which elaborate the TZPs, express EGFR but not LHCGR[19,28,29]. This suggested to us that TZP loss might be mediated through an EGFR-regulated pathway. To test whether EGFR signaling drives the loss of TZPs during maturation, we incubated COCs in the presence of EGF (Fig. 2c, d). After 4 h

of incubation, the oocytes had undergone GVBD and contained condensed chromosomes, as observed during maturation in vivo. At this time, $87.4 \pm 9.5\%$ ($n = 34$) of the starting population of TZPs remained. By 8 h, many of the oocytes had completed the first meiotic division, as indicated by the presence of the first polar body (arrow in Fig. 2c, EGF 8 h). Importantly, only $8.6 \pm 6.1\%$ ($n = 39$) of the TZPs remained, closely matching the results observed in vivo. In contrast, COCs incubated for 8 h in the absence of EGF retained $74.2 \pm 15.3\%$ ($n = 55$) of the initial TZP population. To verify that EGF-triggered TZP loss through

activation of its receptor, we incubated EGF-treated complexes in the presence of either of two inhibitors of EGFR signaling, PD153035 and AG1478, previously used on mouse ovarian follicles in vitro[30]. Both effectively suppressed the EGF-triggered loss of TZPs (Supplementary Fig. 1a). A small decrease in TZP number occurred even in the presence of the EGFR inhibitors, and we address this EGFR-independent decrease later in this report. These results show that activation of EGFR signaling in COCs in vitro triggers loss of the TZPs and that the timing of this process recapitulates that observed in vivo.

Because phalloidin stains F-actin, the presence or absence of staining could not establish whether the membranous structure of the TZP, as well as its actin backbone, was lost. To meet this concern, we evaluated two markers associated with the plasma membrane. We stained COCs obtained from wild-type mice using anti-β-catenin, and COCs obtained from (Gt(ROSA) 26Sor[tm4(ACTB-tdTomato,-EGFP)Luo]) that express a membrane-targeted Tomato Red using anti-RFP (red fluorescent protein). Because of the weak fluorescent signal obtained using anti-RFP, we removed the bodies of the granulosa cells prior to fixation, leaving the TZPs embedded within the zona pellucida. Prior to maturation, both markers revealed the dense TZP network also observed using phalloidin (Fig. 2e). In contrast, 8 h after exposure to EGF, very few TZPs were detectable using either membrane marker. These observations establish that, both in vivo and in vitro, the TZPs are synchronously lost 4–8 h following the initiation of maturation.

The loss of TZPs could be an active process—due to retraction of the filopodial structure—or passive—due to the displacement of the cumulus cells away from the oocyte during maturation (see Fig. 1). Careful examination of COCs between 4 and 8 h following administration of hCG (in vivo) or EGF (in vitro) revealed that the layers of cumulus granulosa cells nearest to the oocyte had not become displaced, but instead remained adjacent to the zona pellucida (Fig. 2f). Short projections often extended from these cells into the zona pellucida. When multiple projections extended from a single locus at the cell surface, the bases of these projections typically were clustered, forming a bouquet-like structure (Fig. 2f, arrowheads). These observations indicate that, following EGF stimulation, the TZPs shorten in concert with a reorganization of the cortical actin network of the cumulus granulosa cells. We observed the same loss of TZPs when we allowed porcine COCs to undergo maturation in vitro, where the multiple layers of cumulus cells remain closely adjacent to the oocyte (Supplementary Fig. 1b). These results definitively establish that the loss of TZPs from the cumulus cells adjacent to the oocyte during maturation is due to an active process of retraction.

**Cyclic GMP stabilizes TZPs in COCs.** Previous reports have shown that, when COCs are incubated in vitro in the absence of EGFR ligands, the oocytes mature and the TZPs are lost[11,18]. To understand the relationship between these observations and our results, we incubated COCs for an extended period of time in the absence of EGF (Fig. 3a, b). Consistent with the previous reports, the oocytes within the COCs underwent maturation, as indicated by the presence of condensed chromosomes at 4 h and 8 h (arrows in Fig. 3a) and of the first polar body at 12 h and 16 h. After 8 h of incubation, 70.7 ± 14.8% (n = 54) of the original population of TZPs remained, indicating that a modest decline had occurred by this time. After 16 h of incubation, however, only 2.6 ± 1.9% (n = 16) remained. These results indicate that TZPs are lost in vitro even in the absence of EGF, but the timing is considerably delayed compared to in its presence. EGFR signaling requires its dimerization, and this can occur in the absence of

ligand-binding[31], raising the possibility that EGFR in cumulus cells might exhibit a weak ligand-independent activity eventually leading to TZP retraction. In contrast to the EGF-triggered loss of TZPs, however, the slow loss of TZPs in the absence of EGF was not attenuated by the chemical inhibitors of EGFR signaling (Supplementary Fig. 1c). These results indicate that, although EGFR signaling is required to recapitulate the physiological timing and synchrony of TZP retraction observed in vivo, an EFGR-independent mechanism can drive a slow process of retraction in vitro.

Fully grown oocytes within antral follicles are held in meiotic arrest through the action of cyclic GMP (cGMP)[19]. cGMP is produced by the membrane guanylyl cyclase natriuretic peptide receptor 2 (NPR2) upon activation by its ligand, C-type natriuretic peptide (CNP) (Fig. 3c). Whereas NPR2 is present on both mural and cumulus granulosa cells, CNP is secreted only by the mural granulosa cells[32]. Consequently, COCs once removed from the antral follicle and placed in culture become deprived of CNP. This leads to a drop in cGMP in both the cumulus granulosa cells and the oocyte, enabling the latter to initiate meiotic maturation. We wondered whether the drop in cGMP might underlie the slow EGFR-independent TZP loss.

To test this, we incubated COCs in the presence of CNP, which has previously been shown to elevate cGMP within COCs and block maturation[32–34] or 8-pCPT-cGMP, a membrane-permeable analogue of cGMP[35]. Oocytes incubated for 18 h in the presence of either CNP or 8-pCPT-cGMP remained in meiotic arrest, as indicated by the presence of the GV (dashed circles in Fig. 3d), confirming that intracellular cGMP remained elevated. Moreover, the population of TZPs remained remarkably stable (98.2 ± 13.2%, n = 34 for CNP; 98.9 ± 9.0%, n = 28 for 8-pCPT-cGMP) (Fig. 3d, e). The ability of CNP to prevent TZP loss has previously been reported[12,36,37], and our results using 8-pCPT-cGMP provide evidence that as anticipated this effect is mediated through elevated intracellular cGMP.

LH triggers a rapid drop in cGMP concentration in both the cumulus cells and the oocyte via multiple mechanisms including increased activity of the cGMP-specific phosphodiesterase (PDE) 5A[38] and decreased activity of NPR2[39,40], and this drop is detectable within minutes after exposing intact antral follicles to LH in vitro[41]. Because TZP retraction does not occur until between 4 and 8 h later, the drop in cGMP is unlikely to directly drive TZP retraction in vivo. Nevertheless, we wished to learn whether elevated cGMP could impair the EGF-mediated retraction. To test this, we incubated COCs for 8 h in the presence of EGF and either CNP or 8-pCPT-cGMP (Fig. 3f, g). Whereas 26.7 ± 8.4% (n = 20) of the TZPs remained in the presence of EGF alone, 80.2 ± 20.3% (n = 43) remained when 8-pCPT-cGMP was also present. CNP was less efficient at blocking retraction, as 45.7 ± 14.9% (n = 25) of the TZPs remained after 8 h. We also noted that more than half of the oocytes underwent GVBD under this latter condition, which suggests that cGMP levels declined. The reduced ability of CNP to prevent TZP retraction (and maturation) in the presence of EGF, as compared to in its absence (compare Fig. 3e, g), may reflect the fact that EGFR signaling decreases the activity of NPR2[42]. These results suggest that the LH-triggered decrease in cGMP concentration within the cumulus cells may facilitate efficient EGFR-mediated TZP retraction.

**TZP retraction occurs independently of oocyte maturation.** TZP retraction does not begin until several hours after maturing oocytes have undergone germinal vesicle breakdown (GVBD) and entered metaphase of the cell cycle. Intriguingly, cytoskeletal elements of the oocyte cortical region become extensively

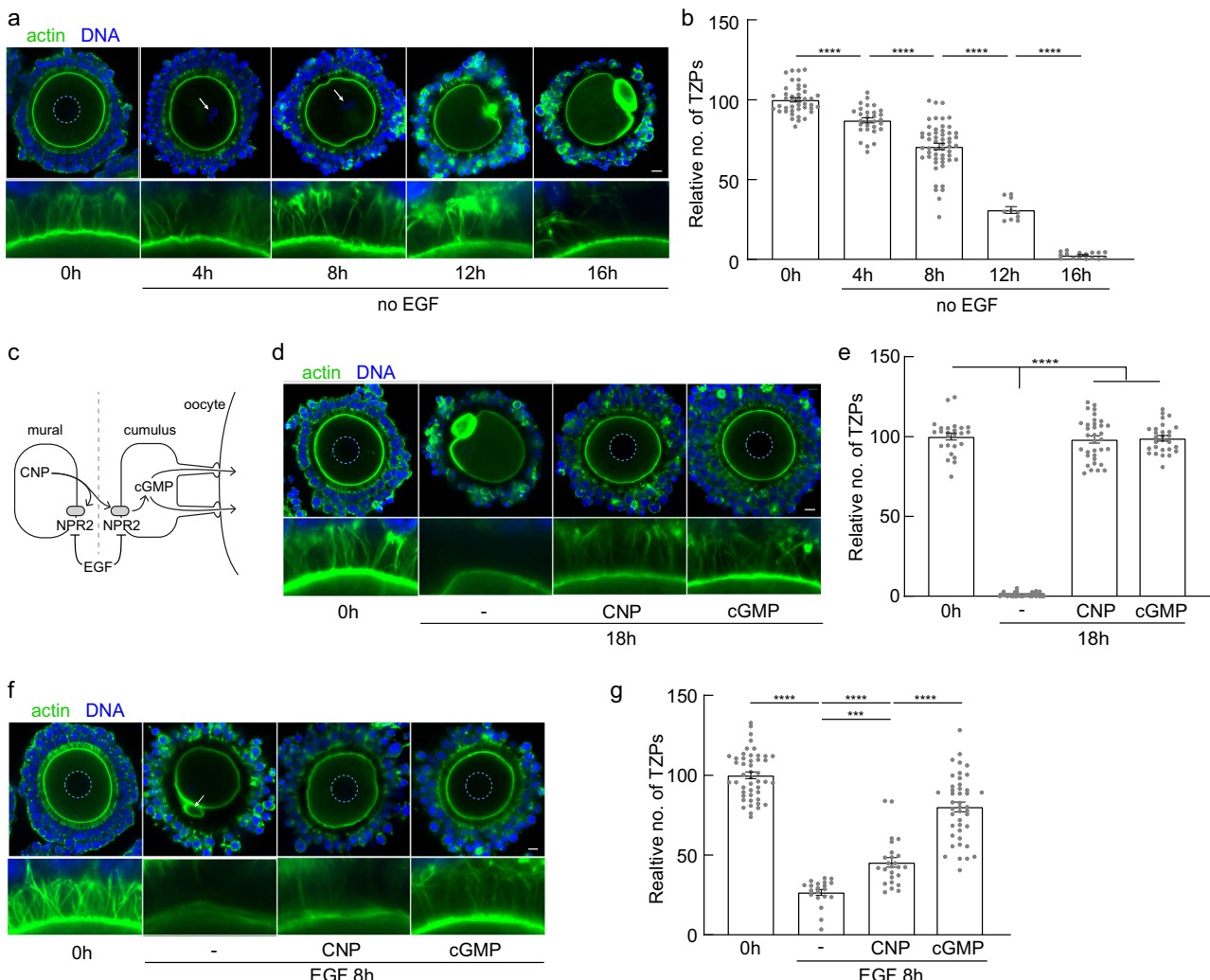

**Fig. 3 Cyclic GMP maintains TZPs. a** Confocal images of COCs obtained from antral follicles of eCG-primed mice and incubated under the indicated conditions. Arrows indicate condensed chromosomes, confirming that GVBD has occurred. **b** Mean and SEM of number of TZPs following incubation under the indicated conditions, normalized to the number at 0 h. Each point represents an individual COC. $n = 42$ (0 h), 30 (4 h), 54 (8 h), 10 (12 h), 16 (16 h) COCs examined over four independent experiments. Statistical analysis using one-way ANOVA with Tukey multiple-comparison test. **** $P < 0.0001$. **c** Schematic showing CNP released by mural granulosa cells activates NPR2 receptor, which produces cGMP, on mural and cumulus granulosa cells. Gap junctions at the TZP tips permit cGMP to be transferred to the oocyte. Gap junctions also connect the mural and cumulus cells. EGF inhibits the activity of NPR2. The dotted line illustrates that COCs removed from antral follicles are deprived of CNP. **d** Confocal images of COCs obtained from antral follicles of eCG-primed mice and incubated under the indicated conditions. Dashed circles outline the nucleus of immature oocytes. **e** Mean and SEM of number of TZPs following incubation under the indicated conditions, normalized to the number at 0 h. Each point represents an individual COC. $n = 25$ (0 h), 31 (18 h), 34 (18 h CNP), 28 (18 h cGMP) COCs examined over three independent experiments. Statistical analysis using one-way ANOVA with Tukey multiple-comparison test. **** $P < 0.0001$. **f** Confocal images of COCs obtained from antral follicles of eCG-primed mice and incubated under the indicated conditions. Dashed circles outline the nucleus of immature oocytes; arrow indicates polar body of mature oocyte. **g** Mean and SEM of number of TZPs following incubation under the indicated conditions, normalized to the number at 0 h. Each point represents an individual COC. $n = 47$ (0 h), 20 (8 h EGF), 25 (8 h EGF CNP), 43 (8 h EGF cGMP) COCs examined over four independent experiments. Statistical analysis using one-way ANOVA with Tukey multiple-comparison test. **** $P < 0.0001$. *** $P = 0.009$. Scale bar (**a**, **d**, **f**) = 10 μm.

modified early during maturation[43,44]. We observed that prior to maturation, E-cadherin is distributed in small foci that are present around the circumference of the oocyte, likely reflecting its localization at the points of membrane contact with the TZPs (Fig. 4a) and consistent with reports implicating cadherins in maintaining adhesion between the granulosa cells and oocyte[45,46]. Following GVBD, however, membrane-associated E-cadherin is no longer detectable, although it reappears at the surface of oocytes that have completed maturation to metaphase II (ref. [47]). β-catenin becomes similarly depleted from the oocyte surface following GVBD (Fig. 4a). The loss of surface E-cadherin together

with the relative timing of GVBD and TZP retraction suggested that changes in the oocyte membrane during maturation might uncouple adhesion with the TZPs, leading to their retraction.

Oocyte maturation begins when the decrease in cGMP relieves inhibition of phosphodiesterase (PDE) 3A within the ooplasm, enabling it to hydrolyze cyclic AMP thereby permitting activation of cyclin-dependent kinase (CDK) 1[19,30,48](Fig. 4b). To test whether changes in the oocyte during maturation underpin TZP retraction, we exposed EGF-treated COCs either to cilostamide, a PDE3A inhibitor that maintains elevated intracellular cAMP, or to roscovitine, which acts downstream by blocking CDK1 activity

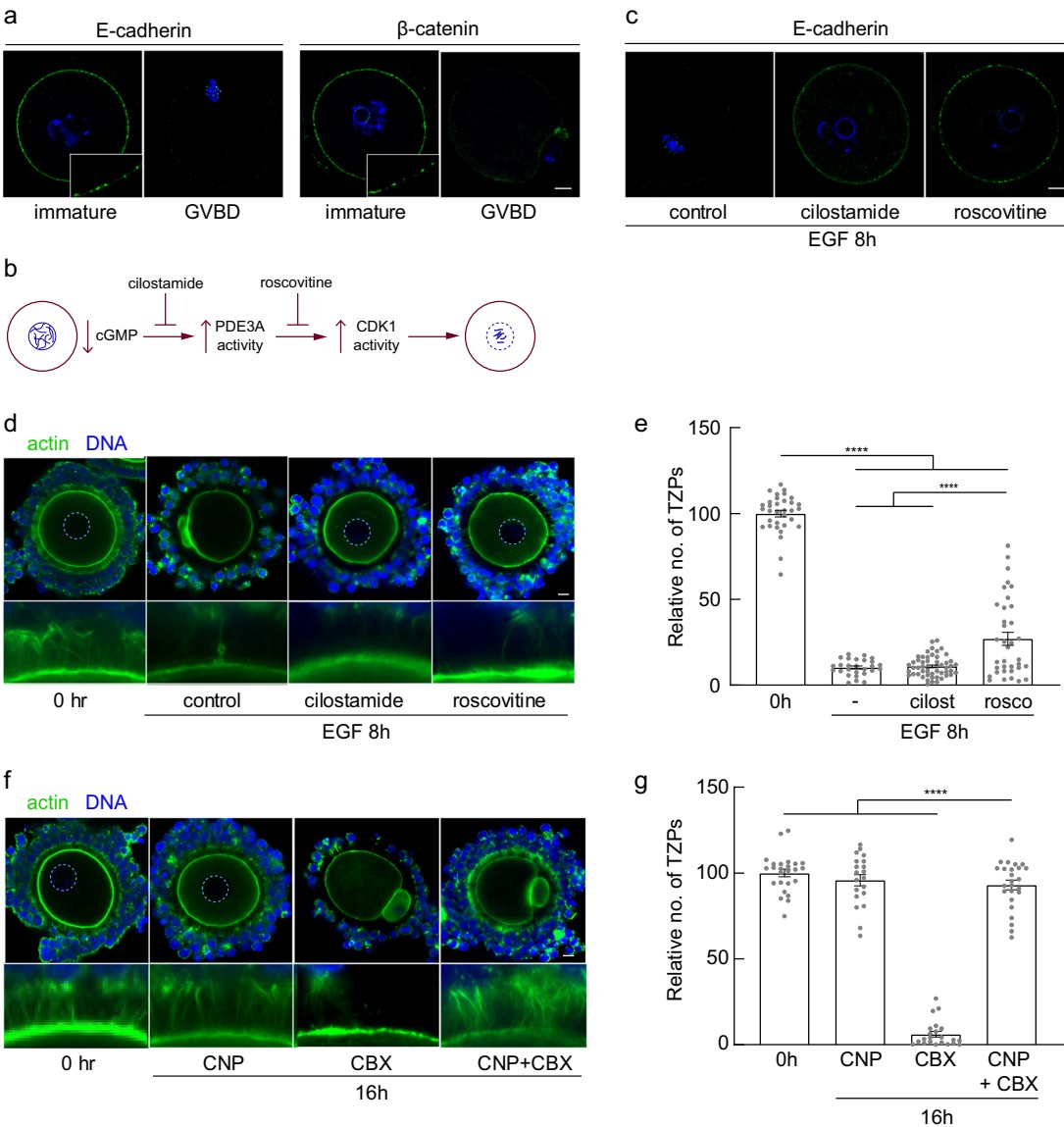

**Fig. 4 TZP retraction occurs independently of oocyte maturation. a** Confocal images of oocytes obtained at the indicated stages and stained using antibodies against E-cadherin or β-catenin. Both stain small foci at the cortex of immature oocytes, but not after GVBD (note oocyte on the right has emitted a polar body). **b** Schematic showing regulation of oocyte maturation. A decrease in cGMP within the oocyte releases PDE3A from inhibition, which by hydrolyzing cAMP enables CDK1 activity to increase, triggering maturation. Cilostamide and roscovitine inhibit the indicated steps in the pathway. **c** Immature oocytes were incubated in the presence of EGF and the indicated inhibitors. E-cadherin foci remain at the cortex. **d** Confocal images of COCs obtained from antral follicles of eCG-primed mice and incubated for 8 h in the presence of 10 ng/ml EGF and the indicated inhibitors. Dashed circle outlines oocyte nucleus. **e** Mean and SEM of number of TZPs following incubation under the indicated conditions, normalized to the number at 0 h. Each point represents an individual COC. $n = 33$ (0 h), 27 (8 h EGF), 52 (8 h EGF cilostamide), 36 (8 h EGF roscovitine) COCs examined over four independent experiments. Statistical analysis using one-way ANOVA with Tukey multiple-comparison test. **** $P < 0.0001$. **f** Confocal images of COCs obtained from antral follicles of eCG-primed mice and incubated for 16 h in the presence of the indicated inhibitors. Dashed circle outlines oocyte nucleus. TZPs remain even when the oocyte has undergone maturation (CNP + CBX). **g** Mean and SEM of number of TZPs following incubation under the indicated conditions, normalized to the number at 0 h. Each point represents an individual COC. $n = 25$ (0 h), 20 (16 h CNP), 22 (16 h CBX), 25 (16 h CNP CBX) COCs examined over three independent experiments. Statistical analysis using one-way ANOVA with Tukey multiple-comparison test. **** $P < 0.0001$. Scale bar (**a, b, d, f**) = 10 μm.

(Fig. 4b). Both inhibitors effectively blocked maturation, as indicated by the presence of the GV, and inhibited the depletion of cortical E-cadherin (Fig. 4c). Despite the meiotic arrest, however, only $11.0 \pm 6.0\%$ ($n = 52$) of the TZPs remained in the COCs exposed to cilostamide, compared to $10.3 \pm 4.6\%$ ($n = 27$) in the control (Fig. 4d, e). Roscovitine modestly suppressed TZP retraction, such that $27.0 \pm 22.8\%$ ($n = 36$) remained of the population present prior to treatment. As roscovitine inhibits the

maturation pathway downstream of cilostamide, the partial suppression of retraction may reflect a non-specific action of the drug. These results demonstrate that TZP retraction is not driven by a mechanism that is activated downstream of the decrease in oocyte cAMP.

To further test whether oocyte maturation controls TZP retraction, we incubated COCs in the presence of CNP while blocking gap junctional activity. This experimental condition

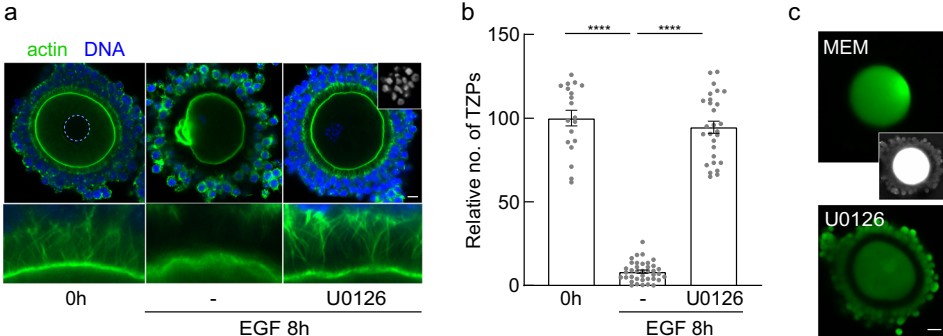

**Fig. 5 TZP retraction requires ERK activity. a** Confocal images of COCs obtained from antral follicles of eCG-primed mice and incubated under the indicated conditions. Dashed circle outlines oocyte nucleus. Inset in the U0126 image shows condensed chromosomes, confirming that GVBD has occurred. **b** Mean and SEM of number of TZPs following incubation under the indicated conditions, normalized to the number at 0 h. Each point represents an individual COC. $n = 19$ (0 h), 38 (8 h EGF), 28 (8 h EGF U0126) COCs examined over three independent experiments. Statistical analysis using one-way ANOVA with Tukey multiple-comparison test. **** $P < 0.0001$. **c** Following an 8 h exposure of COCs to EGF+/− U0126, Lucifer Yellow was injected into the oocyte and fluorescence was visualized 30 min later. Inset is an over-exposure to confirm that cumulus cells are present in the control group. Scale bar = 10 μm (**a**); 20 μm (**c**).

generates high levels of cGMP in the granulosa cells, yet low levels in the oocyte. Under this condition, the oocytes underwent maturation, as expected because cGMP transfer from the granulosa cells had been blocked (Fig. 4f, CNP + CBX, note polar body). Remarkably, however, $93.1 \pm 14.0\%$ ($n = 25$) of the TZPs remained (Fig. 4f, g). These results establish that although TZP retraction and oocyte maturation are temporally linked, they are mechanistically independent. Rather, TZP retraction is regulated primarily or exclusively by mechanisms operating within the granulosa cells.

**TZP retraction requires ERK activity**. We next sought to identify these mechanisms. A key mediator of EGFR signalling in many cell types, including the granulosa cells, is the ERK-MAP kinase (hereafter termed ERK) pathway[33,49,50]. To test whether the ERK pathway mediates EGFR-induced TZP loss, we co-cultured COCs with EGF and U0126, a highly selective inhibitor that has been widely used to block this pathway in oocytes and granulosa cells[51,52]. Consistent with previous reports, oocytes underwent GVBD but failed to complete the first meiotic division in the presence of U0126 (Fig. 5a, inset). Strikingly, $94.6 \pm 19.5\%$ ($n = 28$) of the TZPs remained, indicating that inhibiting ERK activity completely blocked TZP retraction (Fig. 5a, b).

In view of the normal morphology of the TZPs, we then tested whether they retained functional properties. We treated COCs as above and, at the end of the 8-h culture period, injected Lucifer Yellow, a fluorescent dye that can pass through gap junctions, into the oocyte[15,53]. Following a 30-minute incubation to allow dye-transfer, we imaged the COCs. In COCs cultured in EGF alone, no fluorescence was observed in the cumulus cells, indicating that gap junctions no longer connected them to the oocyte (Fig. 5c). In contrast, extensive dye-transfer to the cumulus granulosa cells occurred in COCs cultured in EGF and U0126, indicating that functional gap junctions remained. These results demonstrate that the oocyte and cumulus granulosa cells remain in contact and communication, even in the presence of EGF, when ERK signaling is blocked and provide a mechanistic basis for previous observations that in vivo genetic or in vitro pharmacological inhibition of ERK activity in granulosa cells at the time of ovulation prevents the loss of germ-soma gap junctional communication[49,54].

**TZP retraction requires activity of calpain and Arp2/3**. We then searched for potential targets of ERK signaling. The calpains

are ERK-activated cysteine proteases whose targets include proteins implicated in cell adhesion and whose activation is associated with a loss of actin-based cell protrusions[55,56]. Calpain 2 is expressed in granulosa cells, but not in oocytes, and calpain activity slowly increases during maturation, peaking at 8 h post-hCG[57]. Moreover, calpain is activated in COCs in vitro by EGFR ligands, and its activation is blocked both by U0126 and by a commonly used small-molecule inhibitor, calpain inhibitor I (CI-I)[57]. To test whether activation of calpain was required for TZP retraction, we exposed EGF-treated COCs to CI-I. Consistent with the previous report[57] and confirming the activity of the drug, we observed that expansion of the cumulus layer was impaired. Quantification of the number of TZPs revealed that $56.2 \pm 25.0\%$ ($n = 38$) remained in COCs exposed to EGF and CI-I, as compared to only $12.7 \pm 12.5\%$ ($n = 30$) in COCs exposed to EGF alone (Fig. 6a, b). These TZPs were closely apposed to the oocyte surface and morphologically indistinguishable from those of COCs not treated with EGF. Inhibiting calpain activity thus blocks TZP retraction at an early stage.

The observation that TZPs remained closely apposed to the oocyte surface in the presence of calpain suggested that it may regulate adhesion between the TZPs and the oocyte. Previous studies suggest that adhesion may depend on interaction between N-cadherin and E-cadherin on the surface of the TZPs and oocyte, respectively[45,46]. We found that N-cadherin is present at points of contact between the TZPs and oocyte (Fig. 6c, arrow, asterisks). Notably, in some cases, we detected N-cadherin along the shaft of the TZP (Fig. 6c, arrowhead), confirming its granulosa cell origin. We also detected tight junction protein-1 (TJP1, also known as ZO-1), a component of the protein complex that couples cadherins to actin[58], at the TZP-oocyte interface (Supplementary Fig. 2a; see also ref. [45]). To test whether TZP-oocyte adhesion depends on adherens junctions, which are calcium-dependent, we incubated COCs in Ca-free medium for 15 min and then fixed them. We found that this treatment provoked a widespread detachment of the TZPs from the oocyte surface (Fig. 6d). Although depletion, albeit brief, of external calcium may affect other cellular components, these results support the interpretation that adherens junctions are required to maintain adhesion between the TZPs and the oocytes.

This result suggests that calpain-dependent disruption of adherens junctions may be an early step in TZP retraction. To evaluate this possibility, we treated COCs with EGF in the presence or absence of calpain inhibitor and counted the number of N-cadherin foci, which represent the sites of TZP-oocyte

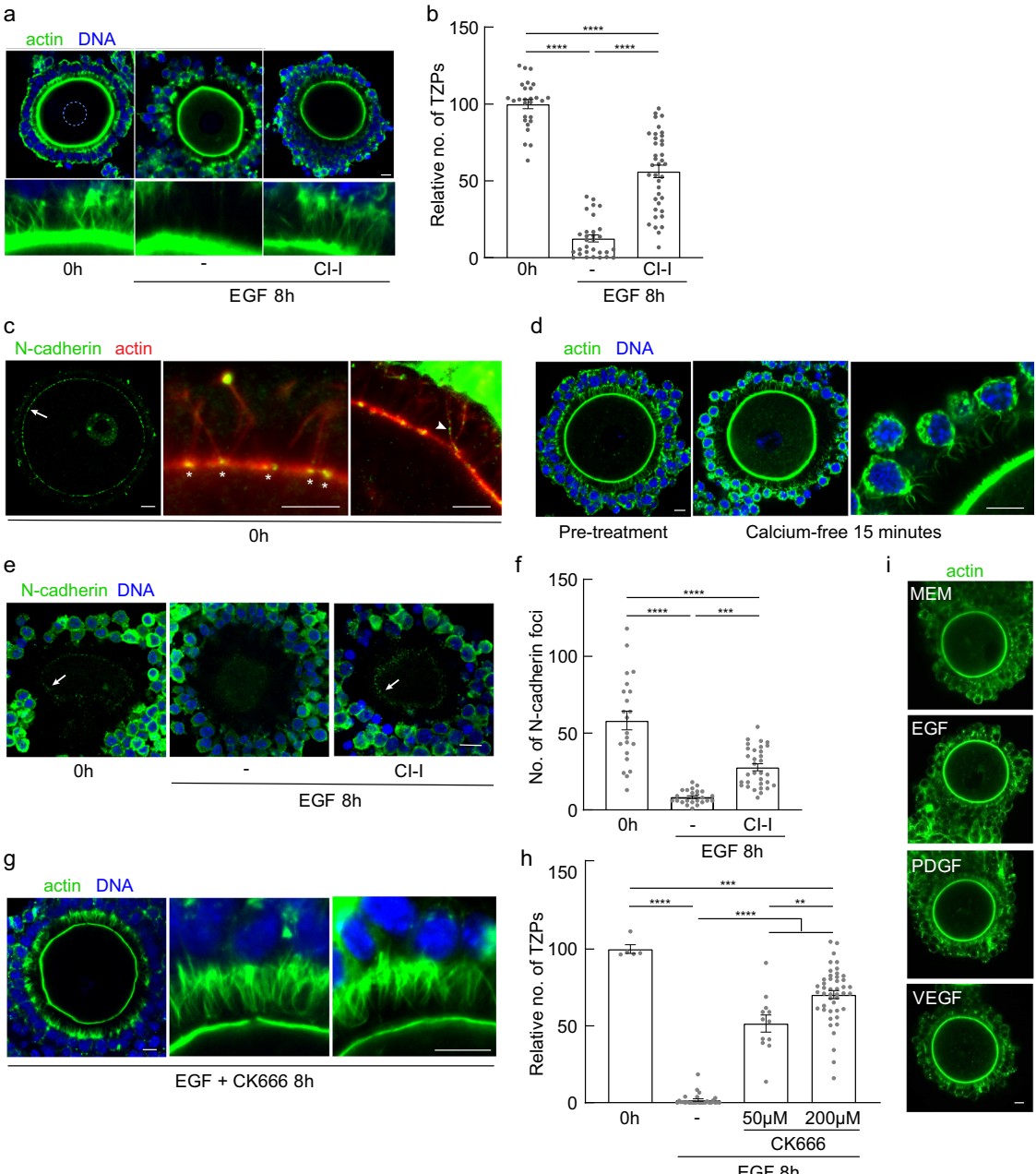

**Fig. 6 TZP retraction requires activity of calpain and Arp2/3. a** Confocal images of COCs from antral follicles of eCG-primed mice, incubated under the indicated conditions. Dashed circle outlines oocyte nucleus. **b** Mean and SEM of number of TZPs following incubation under the indicated conditions, normalized to the number at 0 h. Each point represents an individual COC. $n = 27$ (0 h), 30 (8 h EGF), 38 (8 h EGF CI-I) COCs examined over four independent experiments. Statistical analysis using one-way ANOVA with Tukey multiple-comparison test. ****$P < 0.0001$. **c** Confocal image of COC stained using anti-N-cadherin (green) and phalloidin (red). Arrow shows green foci associated with oocyte surface. Granulosa-cell bodies removed for clarity. High-magnification images show green foci at TZP-oocyte interface (asterisks) and on the TZP axis (arrowhead). **d** Confocal images of COCs before and after exposure to Ca-free medium. **e** Confocal images of COCs from antral follicles of eCG-primed mice, incubated under the indicated conditions and stained using anti-N-cadherin. Images recorded at a plane above the equator to enable N-cadherin foci to be detected. **f** Mean and SEM of number of cadherin foci following incubation under the indicated conditions. Each point represents an individual COC. $n = 22$ (0 h), 25 (8 h EGF), 32 (8 h EGF CI-I) COCs examined over three independent experiments. Statistical analysis using one-way ANOVA with Tukey multiple-comparison test. ****$P < 0.0001$ ***$P = 0.002$. **g** Confocal images of COCs from antral follicles of eCG-primed mice, incubated under the indicated conditions. First and second panels are the same COC; third panel is a different COC. **h** Mean and SEM of number of TZPs following incubation under the indicated conditions, normalized to the number at 0 h. Each point represents an individual COC. $n = 5$ (0 h), 28 (8 h EGF), 12 (8 h EGF 50 μM CK666), 45 (8 h EGF 200 μM CK666) COCs examined over three independent experiments. Statistical analysis using one-way ANOVA with Tukey multiple-comparison test. ****$P < 0.0001$ ***$P = 0.0003$ **$P = 0.0011$. **i** Confocal images of GOCs incubated for 1 h in the indicated growth factors. Scale bar = 10 μm.

contact, at the oocyte surface under these conditions (Fig. 6e, f). Prior to EGF treatment, we observed 58.2 ± 28.4 ($n = 22$) foci of N-cadherin at the oocyte surface. Eight hr after EGF treatment, however, only 8.4 ± 4.3 ($n = 25$) foci were detectable. This result is consistent with the widespread retraction of TZPs that occurs under these conditions. In sharp contrast, 27.8 ± 12.7 ($n = 32$) foci remained in COCs treated in the presence of the calpain inhibitor. It is noteworthy that the inhibitor exerted a quantitatively similar effect on the number of TZPs and the number of N-cadherin foci (compare Fig. 6b, f). Thus, experimental conditions that inhibit TZP retraction also inhibit the loss of N-cadherin from the TZP tips. Further supporting a link between calpain activity and loss of cell-surface N-cadherin, we observed punctae of N-cadherin staining between cumulus cells prior to EGF treatment and, to a lesser extent, in COCs treated with EGF and the calpain inhibitor (Supplementary Fig. 2b, arrows). In contrast, in the COCs treated with EGF alone, N-cadherin was more uniformly distributed in the cortical regions and could also be detected within the cytoplasm from surface of the cumulus cells (Supplementary Fig. 2b, arrowheads). These results support a model in which activation of EGFR signaling leads to a calpain-dependent loss of N-cadherin from TZP tips, thereby physically uncoupling the TZPs from the oocyte.

To pursue the mechanisms that drive TZP retraction following detachment from the oocyte, we focused on the F-actin backbone, as this must be disassembled or otherwise modified for retraction to occur. Actin monomers are assembled into F-actin by nucleator proteins. Proteins of the formin family generate linear filaments; conversely, the Arp2/3 complex generates branched chains by nucleating new filaments that grow out at an angle from existing filaments. Although the relative roles of Arp2/3 and formins in generating filopodia remains to be fully elucidated (reviewed in[59]), inhibitors of formin and Arp2/3 can reduce and increase, respectively, the number of filopodia[60]. Moreover, the formin, DAAM1, is present in TZPs[15], implying that their growth may be supported by the assembly of linear filaments. Because Arp2/3 may oppose filopodial growth under certain conditions, as noted above, and can be activated through ERK signaling[43,61], we reasoned that activation of Arp2/3 might mediate TZP retraction.

To test this idea, we treated COCs with EGF for 8 h in the absence or presence of the canonical inhibitor of Arp2/3-mediated actin assembly, CK666, which acts by stabilizing the inactive form of the complex[62] and has previously been used to inhibit Arp2/3 in oocytes[43,63]. We found that the TZPs detached from the oocyte under these conditions, as indicated by the visible gap between the tips of the TZPs and the oocyte surface marked by phalloidin staining (Fig. 6g). Strikingly, however, although some TZPs had also become shorter, many continued to span most of the width of the zona (Fig. 6g). When we quantified this data, we found that, whereas 0.02 ± 0.04% of the TZPs ($n = 28$) remained in the COCs treated with EGF alone, 51.6 ± 19.3% ($n = 12$) and 70.4 ± 18.0% ($n = 45$) remained in COCs also exposed to CK666 at concentrations of 50 μM and 200 μM, respectively (Fig. 6h). These results identify two phases of TZP retraction—an Arp2/3-independent detachment from the oocyte surface, which does not require Arp2/3 activity, followed by an Arp2/3-dependent shortening of the TZP.

Finally, we determined at what stage of folliculogenesis granulosa cells acquire the ability to initiate TZP retraction. We collected granulosa-cell oocyte complexes (GOCs) from late pre-antral follicles and cultured them overnight with EGF. Because EGFR expression in pre-antral follicles may be insufficient to respond to its ligands[64] we also tested VEGF and PDGF, which also activate the ERK pathway in target cells, and together with their receptors are expressed in granulosa cells[65,66]. Each growth factor increased phosphorylation of ERK in the granulosa cells, indicating that receptor signaling had been activated (Supplementary Fig. 2c). Despite this, none of the growth factors triggered TZP retraction even after overnight culture (Fig. 6i). We conclude that the ability to initiate TZP retraction is acquired by granulosa cells at the pre-antral to antral transition during folliculogenesis.

## Discussion

Loss of adhesion to the somatic cells that support and direct germ-cell differentiation is essential to germ-cell function. In mammalian females, this occurs through a loss of the specialized filopodia known as TZPs that couple cumulus granulosa cells to the oocyte. Here, we have defined the timing of TZP loss and uncovered components of the mechanism that drives this process. TZPs undergo retraction between 4 and 8 h after the ovulatory process is initiated in antral follicles by injection of hCG, which binds to LHCGR. In contrast to the elaboration of the TZPs, which occurs over a period of 2-3 weeks in the mouse[15] and likely several months in humans, their loss is highly synchronized and occurs within a period of only a few hours. This indicates that the activating signal is generated at the same time throughout the cumulus granulosa cell population or is rapidly propagated among cells. Retraction is also triggered by treating COCs obtained from antral follicles with EGF in vitro, suggesting that, in vivo, LHCGR signaling triggers retraction via its well-established activation of EGFR signaling. In addition, whereas the oocyte plays a key role in TZP elaboration, by sending inductive signals to the granulosa cells, it plays little or no role in TZP loss.

Our results suggest a model for TZP retraction comprising three principal events (Fig. 7). First, activation of EGFR leads to phosphorylation and activation of ERK-MAP kinase[49]. Activation of ERK increases calpain activity in cumulus cells[57]. We propose that, second, this increase disrupts complexes between N-cadherin on the cumulus cell surface and E-cadherin on the oocyte surface, thereby physically uncoupling the two cell types. Activation of ERK can also increase the activity of the Arp2/3 complex[43,61]. We propose that, third, Arp2/3 activation changes the cellular environment from one favouring linear F-actin to one favouring branched F-actin. This causes the linear F-actin filaments of the TZPs shorten owing to continuing pointed-end disassembly[67], leading ultimately to resorption of the TZPs into the body of the cumulus cells.

Our results identify several questions to be addressed in future studies. First, the molecular target of calpain is not yet known. N-cadherin, which is a known substrate of calpain[56], is a potential candidate. Our immunofluorescence results indicate, however, that although membrane-associated N-cadherin foci are lost in a calpain-dependent manner during retraction, the protein remains present in the cytoplasm. This suggests that N-cadherin is not extensively degraded within the time-period of our experiments, although does not exclude a quantitatively minor degradation that is functionally important. Intracellular proteins that interact with the cadherins to regulate the stability of adherens junctions are also potential targets; these include β-catenin, which has been identified as a calpain target[56], as well as α- and p120-catenin, and TJP1, which can bind to actin thereby linking it to cadherins. Alternatively, tight-junction or other complexes may mediate TZP-oocyte adhesion and could be targets for calpain-dependent cleavage; however, these have not been detected at this interface[45]. Second, the mechanism by which EGFR-ERK signaling increases Arp2/3 activity remains to be identified. Arp2/3 is activated by members of the WASP (Wiscott-Aldrich syndrome protein) or WAVE (WASP-family verprolin-homologous

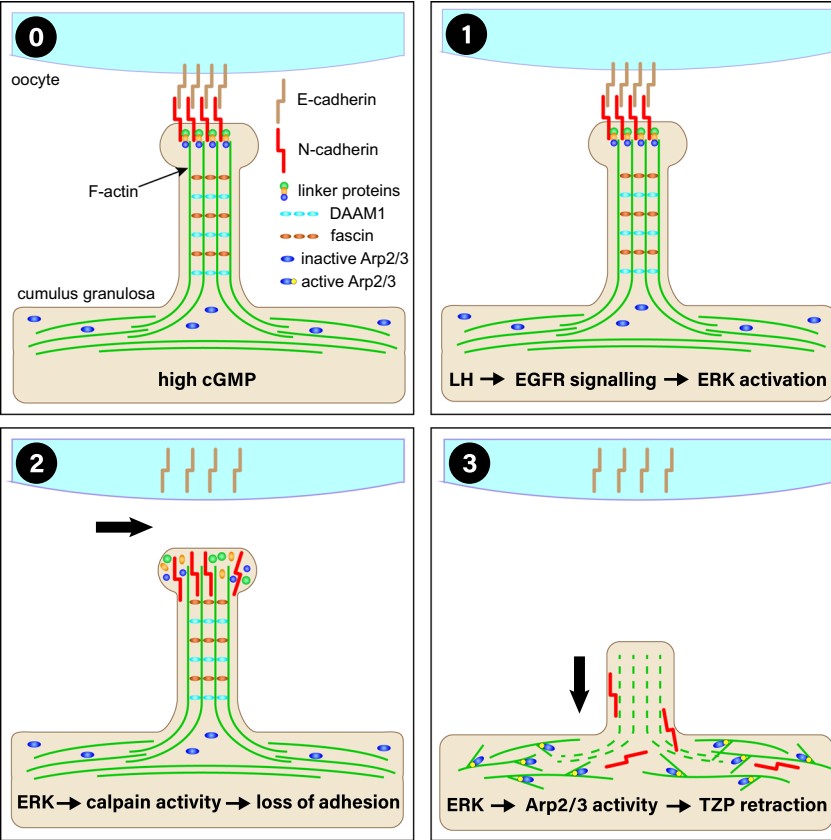

**Fig. 7 Model of TZP retraction.** In antral follicles (stage 0), TZPs projecting from the cumulus granulosa cells to the oocyte contain a backbone of linear F-actin together with the formin, DAAM1, and fascin[15]. N-cadherin (cumulus granulosa) and E-cadherin (oocyte) mediate adhesion between the two cell types. cGMP in the granulosa cells may maintain the TZPs. At the time of ovulation (stage 1), binding of LH to its receptor in the mural granulosa cells triggers release of EGFR ligands, which in turn activates ERK MAP kinase. Active ERK activates calpain (stage 2), which degrades proteins required to maintain N-cadherin at the cumulus granulosa cell membrane, thereby permanently breaking germ-soma contact. Increased activity of the branched chain actin nucleator complex, Arp2/3, which may also be regulated by ERK MAP kinase, reorganizes the actin cytoskeleton (stage 3) leading to a retraction of the TZPs into the granulosa cell body.

protein) families, which by binding to the complex induce a conformational change[68,69]. Thus, it is possible that increased expression or availability of a WASP/WAVE family member triggers an increase in Arp2/3 activity.

Although EGFR signaling drives TZP retraction within the physiological time-frame, TZPs also retract slowly when COCs are removed from the follicle and incubated in the absence of EGFR ligands. This retraction is prevented by agents that raise intracellular cGMP, suggesting that cGMP inhibits an EGFR-independent mechanism of retraction. Although the role of cGMP in regulating oocyte maturation by inhibiting PDE3A in the oocyte is well defined[19], other roles for the cGMP produced by the granulosa cells had not previously been described. The ability of cGMP to inhibit TZP retraction may be relevant to our observation that EGF was unable to induce TZP retraction in pre-antral follicles, even though ERK became phosphorylated indicating activation of receptor signaling. One explanation for this result is that the concentration of cGMP in pre-antral follicles is high and remains so when the GOC is removed from the follicle. Alternatively, pre-antral granulosa cells may lack the machinery to initiate retraction in response to activated ERK. Regardless of the mechanism, the acquisition of the ability to undergo TZP retraction only at the antral stage of folliculogenesis serves to ensure that coupling between the oocyte and its somatic environment is not prematurely terminated. By defining mechanisms that control the developmentally regulated maintenance and

disruption of germ-soma communication, our results may help to refine methods to achieve germ cell development in vitro, both to better understand the molecular basis of cellular totipotency and to preserve fertility in rare species and in women[24,70,71].

## Methods

**Animals**. All experiments were performed in compliance with the regulations and policies of the Canadian Council on Animal Care and were approved by the Animal Care Committee of the Research Institute of the McGill University Health Centre (RI-MUHC; permit #7783). CD-1 mice were obtained from Charles River (St-Constant, QC). mTmG (membrane-Tomato/membrane-Green) founder mice were obtained (Jackson Laboratory, Bar Harbor, ME; strain 007676) and a colony established at the RI-MUHC. These mice carry a transgene encoding a membrane-targeted fluorescent protein. Targeting is mediated via the first 8 amino acids of the plasma membrane associated protein, MARCKS. Mice were housed at 21 °C with 40–60% relative humidity under a 12 h/12 h light/dark regime. Swine ovaries were obtained from prepubertal gilts collected from a local abattoir (Olymel, Saint Esprit, Quebec, Canada).

**Collection and culture of cells**. Murine COCs containing fully grown oocytes and GOCs containing growing oocytes were collected from post-natal day (PD) 19–21 mice that had received an intraperitoneal injection of 5 IU of equine chorionic gonadotropin (Sigma, Windsor, ON) 44 hr previously (referred to as eCG-primed) and PD 17 mice, respectively, as described[22,23]. COCs were collected in Hepes-buffered minimal essential medium (MEM-H) containing cilostamide to prevent meiotic resumption. To permit meiotic resumption, COCs were incubated in cilostamide-free bicarbonate-buffered MEM at 37° in an atmosphere of 5% $CO_2$ in air. Depending on the experiment, the following supplements were added to the indicated final concentration: EGF (10 ng/ml, BD Biosciences 354052), AG1478 (500 nM, Calbiochem 658552) PD153035 (5 µM Calbiochem 234490), CNP

(100 nM, Sigma N8768), 8-pCPT-cGMP (2.5 mM, Sigma C5438), carbenoxolone (150 μM, Sigma C4790), cilostamide (15 μM, Sigma C7971), roscovitine (10 μM, Sigma R7772), U0126 (25 μM, Sigma U120), CI-I (5 μM, Sigma A6185), CK-666 (50 or 200 μM, Sigma SML0006). For experiments involving CI-I and CK666, cilostamide was also added to the culture medium to prevent GVBD so that EGF-triggered retraction could be studied independently of oocyte maturation. GOCs were incubated in the presence or absence of EGF, VEGF (150 ng/mL, R&D 220-BB-010), or PDGF (50 ng/mL, R&D 293-VE-010).

Porcine COCs were collected from 3 to 6 mm follicles and those having at least three layers of cumulus cells and homogeneous cytoplasm were selected for maturation. Ovaries of prepubertal gilts were obtained from a local abattoir (Olymel S.E.C./L.P., Saint-Esprit, QC) and transported to the laboratory in 0.9% NaCl at 32 °C to 35 °C. Follicles ranging from 3 to 6 mm in diameter were aspirated using an 18-gauge needle. Only cumulus-oocyte complexes (COCs) surrounded by a minimum of three cumulus cell layers and having an evenly granulated cytoplasm were selected for IVM. To permit maturation, groups of 30 COCs were incubated at 38.5 °C in a humidified atmosphere of 5% $CO_2$ in air 0.1 ml drops of medium under mineral oil. Initial incubation was in TCM 199 (Life Technologies, Burlington, ON) supplemented with 20% porcine follicular fluid, 1 mM dibutyryl cyclic adenosine monophosphate (dbcAMP), 0.1 mg/ml cysteine, 10 ng/ml epidermal growth factor (Life Technologies), 0.91 mM sodium pyruvate, 3.05 mM D-glucose, 0.5 μg/ml luteinizing hormone (LH; Sioux Biochemical, Inc., Sioux Center, IA), 0.5 μg/ml follicle-stimulating hormone (FSH; Sioux Biochemical), and 20 μg/ml gentamicin (Life Technologies). After 22–24 h of incubation, COCs were transferred to the same medium lacking LH, FSH, and dbcAMP and incubated for an additional 20–22 h.

**Immunofluorescence**. COCs, GOCs, and denuded oocytes were fixed for 15 min in freshly prepared 2% (w/v) para-formaldehyde (Fisher Scientific 04042) in phosphate-buffered saline (PBS, pH 7.2) containing 0.1% Triton X-100 (PBST; ACROS 9002-93-1), then washed in PBST. The specimens were incubated overnight in primary antibody in PBST at 4 °C with gentle agitation, washed twice in PBST, then incubated for 1 hr at room temperature in the secondary antibody as well as phalloidin and DAPI in PBST, then washed in PBST. To mount the samples, a 9 × 0.12 mm spacer (GBL654008, Sigma) was attached to a glass microscope slide. A 2 μl drop of PBS containing 0.3% Polyvinylpyrrolidone (Sigma, PVP360) was placed in the center of the spacer and covered with 20 μl of mineral oil. Samples were then transferred into the drop of PBS and a cover slip was placed on top. Samples were imaged using a LSM 880 confocal microscope (Zeiss, Toronto, ON). The following reagents were used for fluorescent detection of specific proteins: Phalloidin-TRITC (1:100, P1951, Sigma), Phalloidin-Alexa 488 (1:100, A12379, Thermo Fisher), DAPI (1:100, Roche 10236276001), anti-E-cadherin (1:100, BD Transduction Laboratories, 610181), anti-β-catenin (1:100, BD Transduction Laboratories, 610153), anti-RFP (1: 400, Cedarlane, 600-401-379), anti-N-cadherin (1:100, Abcam 18203), anti-TJP1 (1:100, Novus NBP1-85047), anti-rabbit IgG-Alexa 488 (1:100, Thermo Fisher, A11008), anti-mouse IgG-Alexa 488 (1:100, Thermo Fisher, A11001). Tyramide SuperBoost™ Kit with Alexa Fluor™ Tyramide (Invitrogen) was used to amplify the β-catenin signal, following the manufacturer's directions.

**Immunoblotting**. Granulosa cells were collected and lysed in 10 μl of Laemmli buffer. After denaturation, proteins were separated using pre-cast 12% gels (456-8045, Bio Rad) and transferred onto a polyvinylidene fluoride membrane (Amersham, Oakville, ON, Canada) under constant voltage. The membrane was subsequently blocked using 5% non-fat milk, in 0.1% Tween-PBS. The membrane was washed three times in Tween-PBS and incubated with primary antibody overnight at 4 °C. After washing, the membrane was incubated in secondary antibody conjugated to horseradish peroxidase (Promega) at a dilution of 1:5000 for 1 h at room temperature. After the final washes, bound antibody was revealed using ECL+ n (Amersham). Antibodies used were against phospho-p44/42 ERK (1:1000, Cell Signalling 9106) and TACC3 (1:1000, Abcam 134154).

**Confocal image analysis**. To quantify the number of actin-TZPs, a confocal optical section was obtained at the equatorial plane of the oocyte. Using Fiji software (National Institutes of Health, Bethesda, MD), a segmented circle was created around the oocyte circumference, in the middle of the zona pellucida, and the fluorescence intensity at each point on the line was determined. Each point whose value was above the background value of the oocyte cytoplasm and higher than each of its immediately neighboring points was counted as a TZP. The algorithm used to perform this calculation is provided in the Source File. The total number of TZPs counted by this method, although not representing the total number of TZPs in the specimen, was considered as the TZP number associated with the oocyte. For each experiment, except as below, specimens were collected, imaged, and analyzed for each treatment group, and the results were normalized to the mean value obtained for the control group. When the time-course of TZP loss in the absence of EGF was assessed, it was not possible to collect all treatment groups for each experiment.

To quantify the number of N-cad foci, images of the COCs were captured at a z-plane where the oocyte diameter was 50 μm, as the signal at the equator was weak owing to the presence of the cumulus cells. Using ImageJ, the oocyte cortex was traced using Freehand selections tool, and the number of foci of fluorescence above a threshold set at maximal intensity in oocyte cytoplasm were counted in this area. This value was divided by the size area to enable different specimens to be compared.

**Analysis of gap junctional coupling**. The oocyte of each COC was injected with about 10 pl of a 100 mM solution of Lucifer Yellow (Thermo Fisher L12926), using a Zeiss Axio Observer Z1 microscope and PLI-100 microinjector (Medical Systems, NY). The complexes were incubated for 30 min to allow dye transfer to the cumulus granulosa cells and then examined using fluorescence microscopy.

**Statistics and reproducibility**. Data were collected from a minimum of three biological replicates for each experimental series. No technical replicates were included in the data or n-value used for statistical analyses and no data was eliminated from the analyses. Quantitative data were analyzed using GraphPad Prism (9.0.0) employing one-way ANOVA and the Tukey multiple-comparison tests, as described in the figure legends. The confocal images displayed in each figure were obtained from the same experiments that were used to obtain the quantitative data.

**Reporting summary**. Further information on research design is available in the Nature Research Reporting Summary linked to this article.

## Data availability
Data related to the findings of this study as well as the algorithm used to calculate the relative number of TZPs are provided as a source file. Additional information and materials are available from the corresponding author upon request. Source data are provided with this paper.

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

## Acknowledgements

This work was supported by grants from the Canadian Institutes of Health Research (CIHR; PJT 153122), and the Natural Sciences and Engineering Research Council of Canada (RGPIN-402138) to H.J.C. We gratefully acknowledge the generous fellowship support provided by the following agencies: Faculty of Medicine, McGill University (L.A.), McGill Centre for Research in Reproduction and Development (L.A.), CIHR Training Program in Reproduction, Early Development, and the Impact on Health (S.E.H.), Réseau Québécois en Reproduction (S.E.H., S.G.A.), Research Institute of the McGill University Health Centre (S.E.H.), Samuel Solomon Endocrinology Fellowship (K.F.C.), and the China Scholarship Council (W.W.). We thank Shibo Feng and Min Fu (RI-MUHC Imaging Platform) for their invaluable assistance and our colleagues for advice and discussion.

## Author contributions

The project was conceived by S.E.H., H.C., and L.A. Experimental work was carried out by L.A., S.E.H., K.F.C., W.W. Q. Y., and S.G.A. Samples were provided by R.M and V.B. Project supervision and funding were the responsibility of H.C.

## Competing interests

The authors declare no competing interests.
