## [Peer Review File · Nature Communications]

REVIEWER COMMENTS

Reviewer #1 (Remarks to the Author):

When eggs are ovulated, Abbassi et al., report that stimulation of EGFR in granulosa cells triggers synchronized retraction of somatic filopodia that is independent of the egg and requires decreased cGMP as well as ERK activation of calpain.

This carefully conducted research is well described in a manuscript that provides insight into a critical biological process of ovulation with translational implications for human and animal reproduction. Issues that the authors should address:

1. Figure 1c. The authors indicate that activation of EGFR signaling with EGF induces retraction of TZPs. Is TZPs retraction compromised by treatment with EGFR signaling inhibitors or knockdown of cumulus granulosa cell EGFRs in the presence of EGF?
2. Figure 2d. Will in vitro treatment of cultured COCs solely with EGFR signaling inhibitors impair or postpone TZPs retraction?
3. Figure 3. Will treatment only with the cAMP analogue mimic the effect of cilostamide or roscovitine on TZP retraction?
4. In the Discussion, several potential molecular mechanisms that could lead to the shortening of the TZPs are presented. Can they be tested under these experimental conditions in Results to confirm the implicated pathways?
5. Please provide justification for inclusion of porcine COC (lines 138-143) for these mouse studies.
6. Inhibitors can have off-target effects. Can the authors comment on the specificity of the PDE inhibitors that they use to distinguish between somatic and oocyte perturbations of cGMP levels?

Minor points:

1. Line 107, please provide reference for the in vitro method and indicate the COCs are mouse.
2. Reference formats in Literature Cited should be consistent.

Reviewer #2 (Remarks to the Author):

This study addresses the mechanism of germ-soma uncoupling at ovulation. Using confocal microscopy, the authors show that between 4 and 8 hours after LH receptor stimulation in vivo or EGF receptor stimulation in vitro, cumulus cell filopodia retract. Images demonstrating this process are convincing, and show that the retraction cannot be explained by cumulus expansion. This is a new and important finding. However, the mechanistic studies of how this process occurs are less compelling. In particular, the conclusion that a decrease in cGMP is responsible for this response is not well supported. At the concentration used (100 μ M), sildenafil will inhibit many different cyclic nucleotide phosphodiesterases, including those that hydrolyze cAMP (see Bischoff, 2004, International Journal of Impotence Research 16:S11). Likewise at the concentration used (2.5 mM), 8-pCPT-cGMP will also stimulate protein kinase A (Butt et al., 1994; European Journal of Pharmacology 269: 265-268). Thus neither of these reagents is specific for cGMP at the concentrations used. Sildenafil should be used at a

1000x lower concentration (100 nM). Only the experiments using CNP are interpretable in terms of cGMP vs cAMP, and the effect seen with CNP is much less (Figure 2). Thus further studies using more appropriate concentrations of the cGMP modulators are needed to be conclusive. Overall, this study includes important new information demonstrating filopodia retraction in response to hormonal stimulation, but the mechanistic conclusions are less compelling.

For figure one, consider showing higher magnification images. Also, consider showing a graph for the hCG experiment like that shown for the EGF experiment.

We thank the reviewers for their insightful comments and helpful suggestions and the Editors for permitting us to undertake revisions. We have added a substantial amount of new data to the manuscript and extensively edited the text to incorporate this new information. Essentially, the new data allows us to define the mechanism of TZP retraction in more detail, identifying a role for cadherins and the actin branched-chain nucleator, Arp2/3, thereby providing a clearer picture to readers and identifying new avenues for future research. Below, we summarize the major additions to the manuscript, as well as some unsuccessful efforts on our part, and then respond specifically to each point raised by the reviewers.

The following new data has been added:

- Fig. 1c. We provide quantification of the number of TZPs during maturation in vivo following injection of eCG followed by hCG. This timing closely matches that observed in vitro (Fig. 1e) which we reported in the original manuscript.
- Fig. 4f We show that N-cadherin is expressed at the tips of TZPs where they contact the oocyte surface.
- Fig. 4g We show that a brief (15-min) exposure to calcium-free medium causes the TZPs to uncouple from the oocyte, implying that TZP-oocyte attachment requires adherens junctions (which are calcium-dependent).
- Fig. 4h, 4i We show that blocking calpain activity blocks the removal of N-cadherin from the TZP tips.
- Fig. 4j, 4k We show that blocking Arp2/3 activity does not block detachment of the TZPs from the oocyte surface, but inhibits their retraction into the bodies of the granulosa cells. This result mechanistically uncouples the loss of TZP-oocyte adhesion from TZP shortening.
- Fig. 5 Using the new data, we are now able to propose a multi-step mechanism for TZP retraction.
- Supplementary Fig. 1a. We show that EGF-triggered retraction of TZPs is blocked by two inhibitors of EGFR signaling.
- Supplementary Fig. 1b. We show that the slow EGF-independent triggered retraction of TZPs is not blocked by the inhibitors of EGFR signaling.
- Supplementary Fig. 2a. We show that TJP1 (ZO-1), which mediates contact between cadherins and actin, is expressed at the TZP-oocyte interface, further supporting a role for adherens junctions in maintaining contact between the cells.
- Supplementary Figure 2b. We show that EGF causes a widespread loss membrane-associated foci of N-cadherin in the cumulus cells and that this relocation is impaired when calpain activity is inhibited.

We also tested whether talin and paxillin, which are expressed in cumulus cells and are targets of calpain, might be implicated in TZP-oocyte attachment. Immunoblotting (two biological replicates) did not reveal a calpain-dependent loss of paxillin by 8 hr post-EGF, although TZP retraction has occurred by this time. We could not reliably detect talin by immunoblotting, nor have we been able to consistently

detect immunofluorescent signals for either protein at the TZP-oocyte interface. Therefore, our current data do not support a role for either protein.

As an unbiased approach to identifying calpain targets, we carried out a proteomic analysis of granulosa cells stripped from oocytes prior to EGF treatment and after an 8-hr exposure to EGF in the presence or absence of the calpain inhibitor. Three biological replicates of each group were independently sequenced. Although this analysis identified some proteins that declined in response to EGF in a calpain-dependent manner, none are evidently implicated in cell adhesion or the cytoskeleton. We believe that the small number of candidates identified by proteome analysis reflects the limited sequencing depth of these experiments, owing to the small amount of material that could be obtained.

We have also attempted to introduce siRNAs or DNA constructs into granulosa cells within cumulus-oocyte (COCs) or granulosa-oocyte complexes (GOCs), but have been unable to find an effective tool. Notably, although a lentivirus carrying a GFP-encoding gene infected virtually 100% of primary granulosa cells that had been plated on tissue-culture dishes, it infected virtually 0% of granulosa cells within. We are aware of only one publication in which granulosa cells in COCs or GOCs have been successfully transfected (Mol Reprod Dev 76, 537 [2009]).

Below, we respond to each point raised by the reviewers. Figure and line numbers refer to the revised 'changes accepted' version of the manuscript.

Reviewer #1

1. Figure 1c. The authors indicate that activation of EGFR signaling with EGF induces retraction of TZPs. Is TZPs retraction compromised by treatment with EGFR signaling inhibitors or knockdown of cumulus granulosa cell EGFRs in the presence of EGF?

We tested two inhibitors of EGFR signaling – AG1478 and PD153035. Both inhibited the loss of TZPs following treatment with EGF (Supplementary Figure 1a, lines 116-120) but did not inhibit the slow independent loss of TZPs that occurs in the absence of EGF (Supplementary Figure 1c, lines 158-164).

As discussed above, we have tried several different techniques to introduce oligonucleotides, RNAi, or DNA constructs into granulosa cells. While these have been successful in our hands using granulosa cells growing as monolayers in tissue culture dishes, they have been completely ineffective using granulosa cells in cumulus-oocyte complexes or granulosa-oocyte complexes.

Another potential option is targeted gene deletion. In our experience using the mTmG mouse as a reporter strain, the Amhr2-Cre transgene (Nat Genet 32, 408 [2002]), which has been used to modify genes in granulosa cells, induces recombination in only a small proportion of the cells. Other promoters have been used to drive Cre expression in granulosa cells, but these become active either too early (*Foxl2*) or too late (*Cyp19*) during differentiation to be useful for our purposes.

2. Figure 2d. Will in vitro treatment of cultured COCs solely with EGFR signaling inhibitors impair or postpone TZPs retraction?

As noted above, AG1478 and PD153035 did not block the slow EGF-independent retraction.

3. Figure 3. Will treatment only with the cAMP analogue mimic the effect of cilostamide or roscovitine on TZP retraction?

We did not perform this experiment because we felt that the results would be difficult to interpret. Because cilostamide targets PDE3, which is expressed in the oocyte but not the granulosa cells, it will elevate cAMP only in the germ cell. cAMP analogues will elevate it in both compartments, so the origin of any effect will be difficult to determine. We also note that LH, which provokes retraction in vivo, raises cAMP in the granulosa cells. The kinetics of changes in cAMP concentration triggered by LH likely differ from those induced by cAMP analogues, but again this underscores how difficult it would be to interpret the results.

4. In the Discussion, several potential molecular mechanisms that could lead to the shortening of the TZPs are presented. Can they be tested under these experimental conditions in Results to confirm the implicated pathways?

We agree that this was a limitation of the original submission. We have now extended the mechanistic aspect of this study in four principal ways. First, we show that N-cadherin is present at the points of contact between TZPs and the oocyte, as well in the shaft of the TZPs confirming that it is expressed in the cumulus cells (Fig. 4f, lines 266-270). Second, we show that a brief incubation on Ca-free medium is sufficient to disrupt TZP-oocyte adhesion, which implicates adherens junctions in this interaction (Fig. 4g, lines 270-275). Third, we show that blocking calpain activity, which we showed in the original manuscript to inhibit TZP retraction, also prevents the removal of N-cadherin from TZP tips (Fig. 4h, i, lines 276-292). Fourth, we show that blocking the activity of the branched-chain actin nucleator, Arp2/3, does not prevent loss of TZP-oocyte adhesion, but does inhibit retraction of the TZPs into the bodies of the cumulus cells (Fig. 4j, k, lines 293-315). These new data mechanistically uncouple loss of adhesion from TZP shortening and have enabled us to propose a multi-step mechanism for retraction (Fig. 5, lines 340-348).

5. Please provide justification for inclusion of porcine COC (lines 138-143) for these mouse studies.

We feel that the porcine images provide clear evidence that TZPs are lost during maturation by a process of retraction rather than displacement of the cumulus cells away from the oocyte. We agree, however, that they are not fully integrated with the rest of the studies, and for this reason have moved them to Supplementary Figure 1b (see also lines 142-144).

6. Inhibitors can have off-target effects. Can the authors comment on the specificity of the PDE inhibitors that they use to distinguish between somatic and oocyte perturbations of cGMP levels?

We agree that this is a concern, and it was raised also by reviewer 2. We have met this concern in two ways. First, we have removed the data using the PDE5 inhibitor, . We have also focused the manuscript much more on the mechanisms acting downstream of EGFR activity, since cGMP concentration falls long before the TZPs retract. We believe, nonetheless, that the results indicating that high cGMP maintains TZPs in COCs, at least in vitro, are robust. Moreover, they are fully consistent with data published by other groups.

Minor points:

1. Line 107, please provide reference for the in vitro method and indicate the COCs are mouse.

This information has been added (line 106).

2. Reference formats in Literature Cited should be consistent.

All references are now correctly formatted.

Reviewer #2

However, the mechanistic studies of how this process occurs are less compelling.

As discussed in response to Reviewer 1, we have extended the mechanistic aspect of this study in four principal ways. We repeat our response here for convenience. First, we show that N-cadherin is present at the points of contact between TZPs and the oocyte, as well in the shaft of the TZPs confirming that it is expressed in the cumulus cells (Fig. 4f, lines 266-270). Second, we show that a brief incubation on Ca-free medium is sufficient to disrupt TZP-oocyte adhesion, which implicates adherens junctions in this interaction (Fig. 4g, lines 270-275). Third, we show that blocking calpain activity, which we showed in the original manuscript to inhibit TZP retraction, also prevents the removal of N-cadherin from TZP tips (Fig. 4h, i, lines 276-292). Fourth, we show that blocking the activity of the branched-chain actin nucleator, Arp2/3, does not prevent loss of TZP-oocyte adhesion, but does inhibit retraction of the TZPs into the bodies of the cumulus cells (Fig. 4j, k, lines 293-315). These new data mechanistically uncouple loss of adhesion from TZP shortening and have enabled us to propose a multi-step mechanism for retraction (Fig. 5, lines 340-348).

In particular, the conclusion that a decrease in cGMP is responsible for this response is not well supported. At the concentration used (100 μ M), sildenafil will inhibit many different cyclic nucleotide phosphodiesterases, including those that hydrolyze cAMP (see Bischoff, 2004, International Journal of Impotence Research 16:S11). Likewise at the concentration used (2.5 mM), 8-pCPT-cGMP will also stimulate protein kinase A (Butt et al., 1994; European Journal of Pharmacology 269: 265-268). Thus neither of these reagents is specific for cGMP at the concentrations used. Sildenafil should be used at a 1000x lower concentration (100 nM). Only the experiments using CNP are interpretable in terms of cGMP vs cAMP, and the effect seen with CNP is much less (Figure 2). Thus further studies using more appropriate concentrations of the cGMP modulators are needed to be conclusive.

As noted in the response to Reviewer 1, we agree that this is a concern and have therefore removed the data concerning the sildenafil. We cannot exclude side-effects of the 8-pCPT-cGMP, but note that its ability to block the slow EGF-independent TZP retraction matches that of CNP and that previous studies also report that raising adding CNP to the culture medium blocks EGFR-independent TZP retraction (refs 12, 36, 37 in the manuscript). We agree that the effects of CNP and 8-pCPT-cGMP in the presence of EGF are difficult to compare. For this reason, in the revised manuscript we do not discuss potential inhibitory effects of cGMP on EGF-mediated retraction, except to note that it remains a possibility (lines 194-195).

For figure one, consider showing higher magnification images. Also, consider showing a graph for the hCG experiment like that shown for the EGF experiment.

Following this suggestion, we have added higher-magnification images to all Figures. We have also quantified the data for the hCG experiment (Fig. 1c, lines 97-101).

REVIEWERS' COMMENTS

Reviewer #1 (Remarks to the Author):

My concerns have been addressed in the revised manuscript.

Reviewer #2 (Remarks to the Author):

The authors have responded well to all of my comments. This paper reports important new findings, and I am happy to recommend it for publication.